# The Time-Resolved Salt Stress Response of *Dunaliella tertiolecta*—A Comprehensive System Biology Perspective

**DOI:** 10.3390/ijms242015374

**Published:** 2023-10-19

**Authors:** Linda Keil, Norbert Mehlmer, Philipp Cavelius, Daniel Garbe, Martina Haack, Manfred Ritz, Dania Awad, Thomas Brück

**Affiliations:** Werner Siemens Chair of Synthetic Biotechnology, Department of Chemistry, Technical University of Munich (TUM), 85748 Garching, Germany; linda.keil@tum.de (L.K.); norbert.mehlmer@tum.de (N.M.); philipp.cavelius@tum.de (P.C.); daniel.garbe@tum.de (D.G.); martina.haack@tum.de (M.H.); manfred.ritz@tum.de (M.R.); dania.awad@tum.de (D.A.)

**Keywords:** *Dunaliella tertiolecta*, genomics, transcriptomics, proteomics, gene prediction, salt stress, salt stress response

## Abstract

Algae-driven processes, such as direct CO_2_ fixation into glycerol, provide new routes for sustainable chemical production in synergy with greenhouse gas mitigation. The marine microalgae *Dunaliella tertiolecta* is reported to accumulate high amounts of intracellular glycerol upon exposure to high salt concentrations. We have conducted a comprehensive, time-resolved systems biology study to decipher the metabolic response of *D. tertiolecta* up to 24 h under continuous light conditions. Initially, due to a lack of reference sequences required for MS/MS-based protein identification, a high-quality draft genome of *D. tertiolecta* was generated. Subsequently, a database was designed by combining the genome with transcriptome data obtained before and after salt stress. This database allowed for detection of differentially expressed proteins and identification of phosphorylated proteins, which are involved in the short- and long-term adaptation to salt stress, respectively. Specifically, in the rapid salt adaptation response, proteins linked to the Ca^2+^ signaling pathway and ion channel proteins were significantly increased. While phosphorylation is key in maintaining ion homeostasis during the rapid adaptation to salt stress, phosphofructokinase is required for long-term adaption. Lacking β-carotene, synthesis under salt stress conditions might be substituted by the redox-sensitive protein CP12. Furthermore, salt stress induces upregulation of Calvin–Benson cycle-related proteins.

## 1. Introduction

Efficient cultivation of algae holds the potential to address greenhouse gas emissions, while simultaneously yielding valuable compounds [1]. For example, the halotolerant microalga *Dunaliella salina* is known for its remarkable natural β-carotene production. ß-carotene production using *D. salina* is implemented industrially, and hence the strain has been biochemically well characterized. However, the same does not hold true for other *Dunaliella* strains. Moreover, the *Dunaliella* genus appears to be genetically and biochemically rather diverse, which calls for a more detailed resolution of the *Dunaliella* genus both on a molecular genetic and biochemical level [2,3,4]. Interestingly, all strains of the *Dunaliella* genus do not possess a rigid polysaccharide cell wall; instead, they are only enclosed by a cytoplasmic membrane [5]. This characteristic enables the algae to quickly adapt to hyperosmotic changes by shrinking the cell’s size but also allows for simple process-centered lytic recovery of intracellular value-adding products, such as ß-carotene. Several members of *Dunaliella* resist high salt concentration up to saturation (around 5.5 M NaCl) [6,7] by the production of glycerol, which acts as an osmolyte preventing cell lysis under osmotic stress conditions [5]. Glycerol production occurs through either photosynthetic CO_2_ fixation or the degradation of starch [8,9] and accumulated glycerol levels can exceed 50% of the cell’s dry weight [10].

The generation of algae-based glycerol through direct carbon capture constitutes a promising technology [10] for the sustainable production of environmentally friendly “green chemicals” [11], such as polyacrylonitrile or its derivative, high-performance “green” carbon fibers [12,13]. The synthesis of glycerol as *Dunaliella* cells’ stress response has been known for several decades [10]. However, the detailed biochemical processes are still poorly understood. Gaining a deeper understanding of the salt tolerance mechanism has multiple benefits. It allows for the identification of metabolic bottlenecks in glycerol production, which could be addressed by process design intervention or genetic engineering to enhance glycerol production [14]. It also aids in the identification of crucial proteins, whose upregulation can increase the yield and tolerance of economically significant crops [5,15].

Several proteomic and transcriptomic analyses have identified important proteins that were upregulated in *D. salina* after salt stress, such as Na^+^/H^+^ antiporter, transferrin, carbonic anhydrases, tubulin, heat-shock proteins, and glycerol-3-phosphate dehydrogenase (G3pdh) [16,17,18,19]. Generally, the most upregulated genes and proteins were related to photosynthesis, amino acid and protein synthesis, carbon fixation, starch hydrolysis, and glycerol synthesis [15,16,20].

Chen et al. [21] already identified the four key enzymes in the glycerol pathway; G3pdh, glycerol-3-phosphate phosphatase (G3pp), dihydroxyacetone reductase (Dhar), and dihydroxyacetone kinase (Dhak). Nevertheless, in their study, only the activity of G3pdh was significantly upregulated in *D. salina*. Therefore, we want to investigate if there are differences in *D. tertiolecta*’s adaption to salt stress.

So far, mainly *D. salina* as a model has been subject to proteomic and transcriptomic studies investigating the intracellular response to salt stress, while for *D. tertiolecta* only transcriptomics studies related to biofuel and nitrogen depletion are available [22,23]. As an additional stress response, *D. salina* demonstrates a heightened synthesis of β-carotene [24]. Many *D. tertiolecta* strains lack this excessive carotene synthesis [25]. However, the absence of carotene biosynthesis in *D. tertiolecta* is still unclear. Therefore, we aimed to identify alternative mechanisms that compensate for the lack of carotene production. Moreover, it has been reported that some *Dunaliella* strains form palmella structures during the stress response, in which the cells accumulate and form clumps [26]. Our working strain of *D. tertiolecta* lacks excessive β-carotene synthesis as well as palmella formation [25]. These physiological and phenotypical variations among *Dunaliella* strains indicate a need for more precise systems biology analysis of each individual strain within the *Dunaliella* genus. Therefore, this study aims to analyze the intracellular response of *D. tertiolecta* to increased salt concentration.

For proteomic analysis, an adequate genome database and specific gene predictions as well as peptide sequences of the proteome are required. However, these were only accessible for other strains like *D. salina* [27] but not for *D. tertiolecta*. Consequently, *in-house* analysis of the *D. tertiolecta* genome was performed using PacBio Sequel IIe sequencer to generate a high-resolution draft genome, which was subsequently annotated in detail [28]. For the generation of a *D. tertiolecta*-specific protein database, the assembled genome was analyzed using mRNA sequence data isolated prior to and after a substantial increase in salt concentration. This database enabled the MS/MS-based identification of differentially expressed *D. tertiolecta* proteins, which allowed quantification and detection of phosphorylated proteins as quick responses to salt stress [29]. Moreover, we were able to identify the upregulation of Ca^2+^ signaling proteins as an early response to salt stress, which is a common feature in higher plants [30]. Additionally, we identified a portfolio of proteins from different biochemical pathways that might be specific to *D. tertiolecta*, as they have not been identified in *D. salina*.

## 2. Results and Discussion

### 2.1. Genome of D. tertiolecta and Functional Prediction of Protein Coding Sequences

The genome of *D. tertiolecta* was sequenced by a PacBio Sequel IIe NGS sequencer resulting in long, high-fidelity (HiFi) sequences with a yield of 7.5 gb of HiFi reads at a quality of >Q20 (Appendix A). The result of the assembly provided a total of 212 MB as primary contigs with 35-fold coverage. To assess the completeness of the genome, BUSCO analysis was performed with the genome sequence versus Eukaryota and Chlorophyta (Figure 1). Although the available datasets are limited to Chlorophyta, more than 86% complete and single-copy BUSCOs could be matched.

Gene models have been generated using Augustus/Augustus BRAKER [31]. For the generation of a comprehensive protein sequence database, which can serve for quantitative proteomic analyses, transcriptome sequence data were exploited. Subsequently, we performed transcriptome sequencing of *D. tertiolecta* without treatment and after exposure to 2 M NaCl for 30 min and 4 h, respectively. Augustus-BRAKER was used to predict genes as well as their amino acid sequences for each transcriptome sequencing using the genome sequence. Furthermore, de novo prediction was performed using the existing Augustus training set for *D. salina* as well as the generated training set of *D. tertiolecta* at the 0 M NaCl control point. The collected data were merged by removing redundant entries to create a comprehensive non-redundant protein database. Overlap in identical protein sequences found in the proteomic data is shown in Appendix A. Functional annotation was performed using DIAMOND against the NR-NCBI protein database [31,32,33,34,35,36,37,38,39].

### 2.2. Proteomic Changes after Salt Shock

Proteomic changes were examined prior to the sudden increase in NaCl concentration (control) as well as 30 min, 4 h, and 24 h after the change. In total, 12,523; 12,381; 12,230; and 14,428 proteins were identified, respectively. Proteins identified after the salt stress were compared to identified proteins before the salt stress (Appendix A). After 30 min, 379 proteins exhibited a significant change in abundance, and after 4 h, 314 proteins. The most significant change was observed after 24 h, with 691 proteins showing a significant difference in abundance. Throughout all three time points, more proteins were upregulated compared to those downregulated (Figure 2), which is in line with previous studies, documenting a higher number of upregulated proteins in other organisms [16,40]. A detailed analysis of protein groups that were up- and downregulated after salt stress is described in the Section 2.3.

It should be noted that highly similar proteins were identified multiple times, as the database is based on transcriptome analysis. If a single gene produces multiple transcripts, all distinct proteins are included in our database. However, the protein group was condensed to a single protein for KEGG and COG analysis, to prevent biases. Furthermore, not all proteins could be annotated. Hence, the following analyses only include proteins that could be successfully annotated.

### 2.3. Cluster of Orthologous Groups Analysis

The Cluster of Orthologous Groups (COG) analysis, shown in Figure 3, confirms a higher number of proteins to be up- rather than downregulated. Furthermore, proteins related to energy production and conversion, amino acid transport and metabolism, and posttranslational modification exhibited the most significant changes in abundance. A substantial upregulation of proteins associated with RNA processing and modification, translation, ribosomal structure, and biogenesis, was observed particularly at 24 h of osmotic change, underpinning a trend already starting 4 h after the change in salt concentration.

Our data are consistent with those of Cheng et al. [41], who described an enhanced abundance of proteins involved in ATP synthesis, protein refolding, carbohydrate metabolism, and transcription in salt-tolerant plants. Additionally, Athar et al. [19] reported upregulation of proteins related to protein translation, protein proteolysis, and energy production.

Compared to cellular process and signaling, and metabolism, there is an increase in the abundance of proteins associated with information storage and processing over time. At 30 min of salt stress, only five proteins within this group were downregulated, and four were upregulated. However, at 24 h of salt stress, a total of 145 proteins exhibited a significant increase in abundance, whereas 31 proteins showed a decrease, indicating the cells adapt to salt increase by stress-associated protein biosynthesis [42] as, particularly, proteins related to translation are enhanced. As salt stress significantly induces the accumulation of misfolded or unfolded proteins in plants [43], it is likely that in addition to the synthesis of stress-related proteins, misfolded proteins need to be repeatedly synthesized in order to maintain their biological function.

Interestingly, the majority of proteins upregulated during multiple time points were upregulated after 30 min and 24 h of salt stress. Most of these proteins could be arranged within four categories: DNA processing and modification (7 proteins), translation (10 proteins), posttranslational modification (6 proteins), and proteins of unknown function (12 proteins). Since proteins in these categories (except for the unknown function) are related to protein biosynthesis, their presence might be necessary to produce new proteins as an immediate response to salt stress. Furthermore, after 24 h, these proteins are likely required again, possibly for the synthesis of proteins associated with long-term adaption [44]. These results align with those of Wang et al. [15], who observed upregulation of proteins related to translation at 3 h and 24 h. However, their data set did not analyze any earlier or intermediate time points.

### 2.4. Induced Expression of Proteins after 30 min Involved in Salt Stress

A galactokinase (11.46-fold upregulation), Snf1-like protein kinase (SnRK1) (5.56-fold upregulation), nuclear pore protein 96 (5.53-fold upregulation), translocation protein Sec62 (4.88-fold upregulation), and a protein with HAT (Half-A-TPR) repeats (7.85-fold upregulation) exhibited upregulation 30 min after exposure to salt stress (Table 1). In this context, the most dysregulated proteins are listed in Table 1 and Table 2.

The most upregulated protein after 30 min of salt stress is a galactokinase, which is the first enzyme of the galactose metabolism and catalyzes the phosphorylation of galactose to galactose-1-phosphate [45]. A study from 2005 suggests that galactose-1-phosphate is involved in RNA metabolism, ribosome biogenesis, and inositol metabolism [46]. However, this hypothesis has not been confirmed. As galactose is present in *Dunaliella* cells, in excess to glucose [47], the enhanced abundance of the galactokinase might be related to the degradation of galactose as an energy source for protein biosynthesis and/or immediate relief of osmotic stress.

Snf1-related protein kinase 1 (SnRK1) is increased 5.56 times and is the plant equivalent of the heterotrimeric AMP-activated protein kinase/sucrose non-fermenting 1 (AMPK/Snf1). In eukaryotes, it controls growth under nutrient-limiting conditions and regulates sugar and energy starvation responses. Additionally, SnRK1 plays a crucial role in controlling developmental plasticity and resilience in plants exposed to various environmental conditions [48,49]. In our observation, algae cells appear to respond to salt stress by initiating protein biosynthesis. Since protein biosynthesis is an energy-intensive process [50], this could cause energy starvation, resulting in increased biosynthesis of SnRK1. Additionally, SnRK1 has been reported to mediate the crosstalk between the stress-related plant hormone abscisic acid (ABA) and salt stress signaling by phosphorylating select signaling components [51,52]. ABA plays an important role in plant stress response and has also been reported in *Dunaliella* [53]. The activation of SnRK1 or ABA signaling leads to similar transcriptional changes, indicating that these stress pathways have shared targets [52].

To the best of our knowledge, there have been no reports of SnRK1 and galactokinase upregulation following salt stress in any *Dunaliella* strain. Considering that most studies have focused on *D. salina*, it raises the possibility that the upregulation of SnRK1 and galactokinase in response to salt stress is a specific adaptation unique to *D. tertiolecta*. However, additional research is required to validate this hypothesis.

The Sec62 protein is located in the membrane of the endoplasmic reticulum (ER). It plays a vital role in regulating cellular calcium homeostasis, intracellular protein translocation, and the compensation of ER stress [54]. Recent studies have revealed that Sec62 serves as an ER-phagy (a newly identified form of autophagy [55]) receptor in mammals during the recovery phase of ER stress. Hu et al. [56] provided evidence that Sec62 in *Arabidopsis* also acts as an ER-phagy receptor in plants. Furthermore, their findings indicate that sec62-mutant plants display heightened sensitivity to salt-induced ER stress, whereas the overexpression of sec62 enhances the ability to tolerate stress. Moreover, they described the involvement of sec62 in the delivery of misfolded or unfolded proteins to the vacuole for degradation [56]. This enzyme has not been observed in *D. salina*’s adaptation to hyperosmotic conditions, indicating that it might be involved in a unique salt stress-induced response in *D. tertiolecta*.

The half-a-tetratricopeptide (HAT) repeat is a helical repeat motif. HAT-repeat-containing proteins are required for RNA processing, as this motif is found in proteins related to RNA metabolism, such as RNA splicing, rRNA biogenesis, and polyadenylation [57,58]. Another protein related to RNA biogenesis is the nuclear pore protein 96, which is involved in mRNA export [59]. The increased amount of these two proteins might support the RNA processing to synthesize new proteins, necessary for adapting to high salt concentrations.

#### Identification of Phosphorylated Proteins under NaCl Stress

Protein phosphorylation is a highly specific and reversible post-translation modification involved in many signal transduction and regulation processes [60,61]. Phosphorylated proteins control various cellular functions, e.g., signal transduction and responses to external stimuli [62]. The phosphorylation can also influence the catalytic activity, stability, and subcellular localization of proteins, but also the interaction with other regulatory components [62,63].

We found seven proteins phosphorylated only at 30 min of salt stress. Five of those could be annotated and were identified as sodium hydrogen exchanger, DEK protein, 26S proteasome non-ATPase regulatory subunit 4 (PSMD4), SIT4 phosphatase-associated protein, and glutamate receptor (GLR).

One of the most important proteins that was phosphorylated after 30 min of salt stress is a sodium hydrogen exchanger, which is responsible for intracellular regulation of pH and Na^+^ concentration [64]. This indicates that the antiporter might play an essential role as a first response to salt stress, maintaining intracellular ion homeostasis [60].

The architectural chromatin protein DEK was also phosphorylated at 30 min of salt stress. Although it does not possess any known enzymatic activity, biochemical studies have demonstrated its DNA-folding properties and its ability to bind histones, DNA, and chromatin [65]. In a biochemical purification and characterization study by Sawatsubashi et al. [66], it was reported that phosphorylated DEK proteins associate with casein kinase 2 (CK2) and serve as a histone chaperone. Considering that the DEK protein became phosphorylated in our study, DEK may contribute to the transcription and translation processes involved in synthesizing new proteins.

The phosphorylated protein PSMD4 belongs to the ubiquitin/26S proteasome pathway, which plays an important role in various aspects of plant biology. In particular, it is involved in plant protein degradation [67]. However, it also plays a role in regulating signal transduction and responses to biotic and abiotic stresses [68,69]. The phosphorylated 26S proteasome consists of two subcomplexes: the core particle (20S), which degrades proteins without ATP hydrolysis [70], and the regulatory particle (19S) [71]. The 19S particle is a receptor that recognizes ubiquitylated proteins and might play a role in their unfolding and translocation into the interior of the 20S [72]. The detected phosphorylation of the PSMD4 in our results might activate the proteasomal activity and thereby might act in control of levels of key proteins, such as regulatory proteins and stress-responsive transcription factors [18,68,69,73]. Furthermore, 26S is responsible for the quality control of proteins and degrades misfolded and denatured proteins [74]. When the 19S binds a ubiquitinated protein, the multiple catalytic sites of the proteasome break down the substrate into short polypeptides, which are further degraded by peptidases into peptides and amino acids that can be recycled by cells [71,72]. As salt stress significantly induces the accumulation of misfolded or unfolded proteins in plants [43], PSMD4 might also play a role in removing these un-functional proteins, which could impair cellular functions.

Additionally, after 30 min of salt stress, the SIT4 phosphatase-associated protein was phosphorylated. In *Saccharomyces cerevisiae*, the *sit4* codes for a Ser/Thr protein phosphatase, a member of the PPP phosphatase family closely related to the PP2A family [75]. SIT4 is involved in regulating the protein kinase C (Pkc1) and the Pkc1-mitogen-activated protein kinase (MAPK) pathway [75,76]. Therefore, the protein plays a vital role in most known biological functions such as cell wall integrity, actin cytoskeleton organization, ribosomal gene transcription, cell proliferation, and growth. These processes are all necessary for the quick adaption to the increase in salt concentration. Furthermore, SIT4 operates downstream of the plasma membrane sensors midline 2 (Mid2), cell wall integrity and stress response component 1 (Wsc1) and Wsc2, and upstream of Pkc1 [75,77], thereby regulating many more downstream processes, which might help the cell to adapt to the high salt concentration. In addition, SIT4 has not been identified as upregulated in other *Dunaliella* strains.

Finally, GLR was phosphorylated after 30 min of salt stress. GLRs are transmembrane proteins of plants and algae, exhibiting ligand-binding and ion channel activity [78,79]. GLRs are linked to environmental signal perceiving, transmission, and plant adaptation to various stresses [80]. Additionally, GLRs are primarily involved in calcium (Ca^2+^) signaling processes within plant cells [81]. These alterations in Ca^2+^ concentration serve in signal transduction to effect various downstream responses involved in the protection of the plant and the adjustment to new environmental conditions [82]. Therefore, GLRs might be important for signal transduction after salt stress. It has been revealed that in *Dunaliella*, the Ca^2+^ concentration increases rapidly after hyper- or hypo-osmotic changes [83], as Ca^2+^ signaling is one of the earliest signaling pathways [84]. In plants and algae, there are four primary classes of calcium-sensing proteins: Ca^2+^-dependent protein kinases (CDPK), calmodulin, calmodulin-like proteins, and calcineurin B-like proteins, along with their interacting kinases [85]. These enzymes play crucial roles in plant signaling and translating calcium signals into various physiological responses by phosphorylating a diverse range of substrates. These targets include ion channels, transcription factors, and metabolic enzymes. Due to the diversity of targets, calcium-sensing proteins have pivotal functions in processes like transcriptional reprogramming, hormonal signaling, and stress tolerance [86].

In this work, the abundance of two calcium/calmodulin-dependent protein kinases was enhanced (30 min: 3.42-fold, 4 h: 2.38-fold and 4.61-fold). The enhancement of these calcium-dependent proteins supports the hypothesis that Ca^2+^ signaling processes are important for the adaption to salt stress. Additionally, the abundance of one calcium-dependent protein kinase significantly decreased after 24 h (0.37-fold). This indicates that calcium-dependent pathways may serve only as a rapid response to salt stress [84]. The Ca^2+^ signaling processes may operate independently from the mitogen-activated protein kinase (MAPK) pathways [87]. Within our genome database, we identified 21 enzymes associated with MAPK, but none exhibited an upregulation at any analyzed time point. Nonetheless, we detected several serine/threonine protein kinases that remain unspecified, which may potentially be assigned as MAPKs. Further research is required to confirm that MAPK indeed does not become upregulated as a response to salt stress and Ca^2+^ signaling processes alone are required for rapid salt adaptation response.

### 2.5. Differentially Expressed Proteins after 4 h of Salt Stress

Interestingly, only minor differential changes in protein expression were detected at 4 h after salt stress initiation. Most proteins with increased abundance at this time point can be categorized into energy metabolism, translation, ribosomal structure, and biogenesis.

Some interesting proteins with elevated abundance after 4 h are a potassium/proton antiporter (3.64-fold) and a mitochondrial transcription termination factor (mTERFs) (7.43-fold) (Table 1). Additionally, CP12 was upregulated 4.25-fold, which is a small, redox-sensitive protein common in most photosynthetic organisms, such as higher plants, cyanobacteria, and green algae [88]. CP12 regulates the Calvin–Benson cycle (CBC) by mediating the formation of a complex between phosphoribulokinase (PRK) and glyceraldehyde-3-phosphate dehydrogenase (GAPDH) [88,89,90]. Moreover, a study conducted by Tamoi et al. [89] demonstrated that CP12 proteins are able to bind NADPH, allowing them to control the flow of electrons from Photosystem I to NADPH [88,89,90]. Thus, CP12 plays a crucial role in protecting cells against oxidative stress and suppressing the production of reactive oxygen species (ROS) [89]. Algae typically generate ROS, which can function as secondary messengers in numerous cellular processes. When exposed to abiotic stresses, the balance between ROS production and suppression is disrupted, usually resulting in an increased ROS level triggering oxidative stress [91].

Notably, CP12 expression was significantly elevated at all measurement time points (30 min: 2.5-fold; 4 h: 4.25-fold; 24 h: 2.48-fold). Although the primary focus of this study was on proteome data, we also examined the corresponding transcriptome data sets for the purpose of generating the database. If there were notable differences between transcriptome and proteomic results, these deviations are noted specifically. In the case of CP12, two different transcripts were identified. One of the corresponding transcripts was enhanced at 30 min of salt stress (1.12-fold). However, the other transcript was decreased at 30 min (0.90-fold) and 4 h (0.56-fold) of salt stress, even though the resulting protein was enhanced over the entire timespan. Since the generation of ROS is a commonly observed phenomenon during salt stress, CP12, as a redox-sensitive protein, might protect the algae cells against ROS. Furthermore, CP12 might play an essential role in the regulation and activation of the CBC, not only as a quick response but also as a long-term adaption to high salt concentrations. Moreover, at 30 min and 24 h of salt stress, other proteins involved in the CBC are also upregulated. These include Fructose-bisphosphatase (30 min: 2.34-fold; 24 h: 2.93-fold), Sedoheptulose-bisphosphatase (30 min: 3.66-fold), Glyceraldehyde 3-phosphate dehydrogenase (30 min: 2.78; 24 h: 1.85), and Ribulose 1,5-bisphosphate carboxylase/oxygenase (Rubisco) (2.95-fold).

Another protein family that is related to the CBC is carbonic anhydrases. These enzymes optimize CO_2_ uptake, play a crucial role in the reversible hydration of CO_2_, and facilitate the conversion of accumulated HCO_3−_ to CO_2_, which is then fixed by RuBisCO [92]. In our study, the gamma carbonic anhydrase 3 was upregulated after 30 min of salt stress (1.75-fold), while another carbonic anhydrase was significantly upregulated (1.87-fold) after 24 h. Liska et al. [93] already reported that three carbonic anhydrases were upregulated in *D. salina* when cultured in 3 M NaCl compared to 0.5 M NaCl. They also observed that carbon uptake activity in cells grown in 3 M NaCl was more than twice as efficient compared to cells grown in 0.5 M NaCl. Additionally, in their study, five major CBC enzymes were upregulated. The upregulation of CBC enzymes and carbonic anhydrases may be a consequence of increased photosynthetic CO_2_ assimilation. As they did not analyze the quick adaption to salt shock, but the long-term adaptation of the algae, these CBC-related proteins might be involved in prolonged adaptions to high salt levels.

When exposed to high salt concentrations, *Dunaliella* accumulates a high amount of glycerol as an osmolyte to balance intracellular osmotic pressure [94]. Glycerol production occurs via photosynthetic CO_2_ fixation or starch degradation [8,9]. Therefore, the detected CBC proteins might be upregulated to enhance CO_2_ binding to supply the cell with glucose for glycerol biosynthesis. Energy carriers generated during CBC could provide the cellular driving force required for salt stress adaption.

Simultaneously, some photosynthesis-related enzymes are upregulated (Table 3). Photosynthesis provides ATP and NADH required for CBC operation. As photosynthetic enzymes are upregulated, and enhanced photosynthesis has been reported in *D. salina*, it could be assumed that the photosynthetic activity in our strain is also enhanced.

Moreover, the significant increase in the Photosystem II nuclear encoded Pbs27, which was elevated ~11 times at 4 h of salt stress, is noteworthy. Pbs27 is mainly responsible for the repair of damaged PSII complexes [95]. Additionally, the study of Nowacyk et al. [96] suggests that Psb27 plays a role in the assembly of the water-splitting site of PSII and in the turnover of the complex. Interestingly, Pbs27 was downregulated on transcriptional level after 30 min (0.72) and 4 h (0.58) of salt stress, but somehow still led to an ~11-fold increase in proteome level, which may be due to reduced protein degradation.

Even though salt stress typically inhibits photosynthesis in most plants and cyanobacteria [93], enhanced photosynthetic CO_2_ assimilation as a response to high NaCl concentration is a common characteristic in *Dunaliella* [15,93]. In general, increased photosynthesis causes the accumulation of ROS, resulting in oxidative stress within cells. This leads to damage in various cellular macromolecules, such as lipids, proteins, DNA, and carbohydrates [97]. *Dunaliella*’s special feature to increase photosynthetic activity under salt stress is connected to the synthesis of antioxidant enzymes and molecules, such as carotene, which helps to maintain the intracellular redox balance and prevent undesirable cellular damage [98]. When *D. salina* enhances photosynthetic activity at high salt concentration, the cells accumulate (β-)carotene to counteract consequent ROS [99]. This might be the most significant distinction between *D. salina* and *D. tertiolecta*, as our *D. tertiolecta* strain neither turns red at high salt concentration nor significantly increases any pigment biosynthesis-related proteins following salt stress. As the carotene synthesis in *D. tertiolecta* was not upregulated after the increase in salt concentration [25], we assume that an alternative mechanism is required to prevent cell damage. Following salt stress, no antioxidative enzymes, such as catalase, peroxidase, or superoxidase [19,97] were significantly upregulated in our study, despite the upregulation of some photosynthesis-related proteins. In contrast to Jahnke et al. [100], who reported enhanced presence and activity of antioxidative enzymes in *D. tertiolecta*, in our study, the amount of proteins involved in ROS and oxidative stress management was not differentially regulated in abundance. Thus, future studies should analyze the activity of these antioxidative enzymes in more detail. Furthermore, it might be possible that our working strain *D. tertiolecta* upregulates CP12, as an alternative route for ROS protection, e.g., by controlling the electron flow from Photosystem I to NADPH [88,89,90]. Moreover, Wang et al. [101] observed that overexpression of SnRK1 results in enhanced ROS metabolism by increased expression of antioxidative genes and antioxidant enzyme activities. In our study, SnRK1 was upregulated after 30 min of salt stress but did not lead to an increase in antioxidant enzyme abundance, which confirms the need for enzyme activity analysis. Moreover, it would be important to quantify the level of ROS within the cells following salt stress. This could provide a more distinct comprehension of the intracellular ROS activity [102].

### 2.6. Differentially Expressed Proteins Related to Long-Term Adaption

After 24 h of salt stress, most enhanced proteins are related to translation, ribosomal structure, and biogenesis (Figure 1). The 12-times upregulated sulfur locus protein 6 (SURF6) is a component of the nucleolar matrix with a remarkable ability to bind nucleic acids [103]. During ribosome biogenesis, this enzyme is involved in different steps of rRNA processing [104]. Therefore, it might support the synthesis of new proteins needed to adapt to high salt concentration.

One of the highest upregulated proteins after 24 h of salt stress was glutamate-dehydrogenase (GDH), which was increased by 23-fold (Table 1). This enzyme catalyzes the reversible deamination of glutamate to α-ketoglutarate and ammonia using NAD(P) as a cofactor [105]. Despite GDH’s ability to assimilate ammonia, recent reports indicate that primarily deamination occurs under stress conditions [106]. GDH is an important regulator of the amino acid metabolism but also improves nitrogen utilization and helps to maintain the carbon–nitrogen balance [107]. The enhanced generation of α-ketoglutarate can be routed into the Krebs cycle, which leads to the synthesis of ATP [108]. Moreover, the GDH pathway is linked to various cellular processes, such as ammonia metabolism, redox homeostasis (via the formation of fumarate), and acid-base equilibrium [109]. However, it remains uncertain whether GDH is upregulated as a result of a necessary cellular nitrogen supply or if it is linked to subsequent downstream biosynthetic processes. This particular question will be addressed in future studies.

The protein with the highest enhanced abundance at 24 h of salt stress is the engulfment and cell motility (ELMO) protein (Table 1). Even though, on the transcription level, the enzyme was downregulated after 4 h of salt stress (0.78-fold), this enzyme was upregulated 32-fold after 24 h of salt stress. ELMO interacts with the cytokinesis proteins Dock180, a guanine nucleotide exchange factor (GEF) of the Rac family, and regulates GEF activity. The ELMO/Dock180/Rac complex regulates cytoskeletal reorganization, cell apoptosis, and cell migration [110,111]. *Dunaliella* cells shrink as a quick response to salt stress [5] and their cell size is restored when an adequate amount of osmolyte is synthesized. Thus, ELMO might have a crucial function in the reorganization of the cell.

Both highly upregulated proteins, GDH and ELMO, have not been identified in other studies, analyzing *D. salina*’s adaption to salt stress. Therefore, these enzymes might be unique to *D. tertiolecta*’s salt stress adaption.

Additionally, a protein with a biotin and thiamin synthesis-associated domain was enhanced 8.97-fold. It has been reported that salt stress leads to the upregulation of thiamine biosynthetic genes in plants [112], which might be the reason for its increased abundance in this work.

### 2.7. Changes in Key Enzymes of the Glycerol Pathway

Our previous results have demonstrated that the glycerol concentration increased 50–60% in cells after salt stress compared to the control, where NaCl concentration was kept constant [113]. This indicates glycerol biosynthesis is a rapid response to generate an osmolyte to prevent cell lysis after salt stress. The glycerol biosynthesis pathway, which is represented in Figure 4, starts with glucose, obtained from photosynthesis or hydrolysis of starch [7], which is primally stored in the chloroplast [114]. Compared to the transcriptomic study of He et al. [16], who observed the upregulation of numerous transcripts for starch-degrading proteins, in our study, none of the enzymes necessary for converting starch into glucose exhibited upregulation in response to salt stress. As we found proteins related to photosynthesis and CBC upregulated, it can be assumed that glucose predominantly originates from photosynthetic processes. The impact of light on *Dunaliella* has already been described by Xu et al. [115], who reported that the cell volume and glycerol content oscillate with the light/dark cycle in *Dunaliella.* This highlights the possible regulatory role of light and indicates that in our study, where algae were cultivated with continuous light, the light could potentially influence cellular processes, e.g., leading to glucose predominantly obtained from photosynthesis.

Furthermore, the 11-fold upregulation of the galactokinase indicates that other sugars such as galactose might also serve as a carbon source in glycerol biosynthesis. Nevertheless, starch-degrading proteins might be activated by other biochemical processes. To precisely determine the source of glucose in *D. tertiolecta*, future investigations should analyze carbon flux by ^14^CO_2_-fixation experiments [116]. These analyses should be performed with continuous light, as well as with light-dark cycles, as the presence or absence of light significantly affects the source of glucose [116].

Glucose is converted to fructose-1,6-diphosphate and then to dihydroxyacetone phosphate (DHAP). In the glycerol synthesis pathway, four key enzymes play a significant role: Glycerol-3-phosphate dehydrogenase (G3pdh) catalyzes the conversion of DHAP into glycerol-3-phosphate (G3P), which is finally converted into glycerol by Glycerol-3-phosphate phosphatase (G3pp). For the degradation of glycerol, glycerol is oxidized to dihydroxyacetone (DHA) by Dihydroxyacetone reductase (Dhar). Finally, the Dihydroxyacetone kinase (Dhak) converts the DHA back to DHAP [7,21].

Since glycerol serves as the primary osmolyte against high salt concentrations, it is expected that the activities of these key enzymes should be modulated. However, in our analysis, only the expression of Dhak was increased after 24 h of salt stress, which suggests upregulated glycerol degradation 24 h after salt stress induction.

In our genomic database, we identified 47 different gene predictions for the G3pdh. Interestingly, we identified two different G3pdh variants. On the one hand, NAD-dependent G3pdhs, responsible for reducing DHAP to G3P, of which one was decreased after 24 h (0.40-fold) of salt stress. On the other hand, mitochondrial G3pdhs (mG3pdh), which belongs to the FAD-dependent G3pdh family, was increased (24 h: 2.64-fold). Additionally, the corresponding transcript of this mG3pdh was increased 1.45-fold after 30 min and 3.82-fold after 4 h of salt stress. This mG3pdh catalyzes the reverse reaction, the oxidation from G3P to DHAP [117]. Wu et al. [118], who characterized the G3pdhs in *D. salina*, identified and characterized seven G3pdh gene families, such as the NAD-dependent G3pdhs, which is located in the chloroplast and cytoplasm and is involved in glycerol synthesis. In our study, the NAD-dependent G3pdhs are also located either in the cytosol or in the chloroplast, as the sequence contained a chloroplast or cytosol signal peptide.

Nevertheless, our database did not enable any identification of G3pp, potentially suggesting its absence in our *D. tertiolecta* strain. G3pp catalyzed the irreversible conversion from G3P to glycerol, representing the commonly known pathways for glycerol synthesis. As we confirm glycerol accumulation in our strain upon salt stress, G3pp, on the one hand, still might be present in our algae, potentially classified as a phosphatase such as “phosphatase domain-containing protein”. This limited annotation is also reported in the genome study of *D. salina* by Polle et al. [27], in which they were also not able to specifically identify any gene prediction for the Dhar but were able to identify several potential alcohol dehydrogenases [24]. On the other hand, the results from He et al. [119] discovered a di-domain G3pdh isoform, which combines the conventional G3pdh domain with a phosphoserine phosphatase-like domain. This chimeric protein catalyzes the two-step conversion of DHAP to glycerol, a process for which most organisms typically require a separate phosphatase protein, as also assumed for *Dunaliella* [21]. In our study, three of our identified G3pdhs also contained the phosphatase-like domain. This indicates that in our *D. tertiolecta* strain, the G3pp indeed might be absent, as the phosphatase-like domain of the chimeric G3pdh substitutes the conversion of G3P to glycerol with no need for an additional phosphatase protein. In future studies, the activity of the G3pdh with a phosphatase-like domain should be analyzed in more detail, to identify which enzyme is responsible for the conversion of G3p to glycerol.

Based on their study, Chen et al. [21] suggested that under hyperosmotic conditions, G3pdh is the rate-limiting enzyme in *Dunaliella*’s glycerol biosynthesis pathway [120]. In their opinion, this enzyme might play a crucial role in regulating the glycerol levels to balance the osmotic pressure induced by salt shock [21]. Contrary to their results, we observed no significant increase in our identified G3pdhs, but even a decrease of one of the NADH-dependent G3pdh at 24 h upon salt stress. As the regulation of the intracellular glycerol concentration is in equilibrium within 2–3 h after osmotic stress [5], and we did not identify any significant changes in abundance of the key enzymes, the enzymes of the glycerol synthesis might be present in sufficient quantities even before changes in extracellular salt concentration occur. However, it still might be possible that these enzymes become activated through other protein modifications [63] including the Ca^2+^ signaling pathways [5,83].

While the glycerol synthesis-related NADH-dependent G3pdh was downregulated, the abundance of mG3pdh was increased. It has been reported that the FAD-dependent mG3pdh is not only responsible for the regulation of cytosolic glycerol and G3P, as a metabolite connecting lipogenesis, oxidative phosphorylation, and glycolysis, but also for the transfer of reducing equivalents from the cytoplasm to the mitochondrial electron transport chain [121,122]. In *Aradopsis*, mG3pdhs are mainly involved in the respiratory chain [121]. Furthermore, it has been reported that mG3pdh plays a crucial role in salt stress tolerance, particularly by maintaining cellular redox homeostasis and ROS balance. This is confirmed by overexpression of mG3pdh in soybean, which reduced cellular damage by oxidative stress [123]. Hence, we assume that the increased presence of mG3pdh is primarily associated with its role as a shuttle for cellular redox control and is not related to glycerol degradation. As we analyzed the amount of proteins but not the functional activity, additional investigations are required to elucidate the precise roles of G3pdh and mG3pdh in the adaptation to high salt concentration by performing (m)G3pdh activity assays in salt-stressed *D. tertiolecta* [124].

Another enzyme that might be a rate-limiting enzyme of glycerol synthesis is phosphofructokinase (PFK), which converts glucose to fructose-1, 6-bisphosphate [5,44]. Some studies reveal that PFK is upregulated under hyperosmotic conditions in *D. salina* [44,125]. In our analysis, the PFK was upregulated 2.26-fold after 24 h of salt stress. The upregulation of the PFK occurs after glycerol synthesis and cell size adaption have been accomplished. This indicates that PFK may not be essential for the immediate adaptation to salt stress but rather plays a role in long-term adaptation to high salt concentration.

### 2.8. Chaperons and Heat-Shock Proteins Affected by Salt Stress

Following salt stress, several chaperones including heat-shock proteins (HSP) were upregulated (Table 4). They play an important role in protein quality control by aiding the proper folding of newly synthesized proteins, facilitating the assembly and disassembly of protein complexes, assisting in the transport of proteins across membranes, breaking down protein aggregates, refolding denatured proteins to their native state, and transferring non-folding proteins to proteolytic degradation [16,111,112]. Most chaperones depend on ATP hydrolysis to promote the correct folding of their substrates and during stress, they prevent aggregation and restore misfolded proteins [126]. As the accumulation of misfolded or unfolded proteins is increased during salt stress, chaperones are required to counteract those that misfold, which leads to higher ATP turnover. This energy consumption might cause energy starvation, which activates further cellular processes, such as the upregulation of photosynthesis or activation of the enzyme SnRK1 (described in Section 2.4). Furthermore, energy starvation in parallel with enhanced photosynthetic activity might lead to the upregulation of ATP synthases in our analysis (chloroplast ATP synthase gamma chain protein (30 min: 1.91-fold) and ATP synthase (24 h: 2.06-fold)), which are able to synthesize ATP by using the energy stored in the transmembrane ion gradient [127].

Recent research has revealed that protein folding is not the only role of chaperones. They also play a crucial role in protein targeting and degradation, which in turn regulates signaling cascades in response to abiotic stresses, such as salt stress [19,128,129].

### 2.9. Differently Expressed Transporters and Transmembrane Proteins

Channels and transporters with altered abundance after salt stress are shown in Table 5. The voltage-dependent anion channel protein 2 was upregulated at 30 min of salt stress. A copper (CU) transporter is upregulated after 4 h and 24 h of salt stress. Therefore, CU might be needed for long-term adaption, since CU is a cofactor in key processes such as photosynthesis, hormone perception, respiration, and protection against oxidative stress [130]. As it is particularly involved in electron transport and free radical scavenging [130], CU might help to protect the cells against oxidative stress caused by salt stress and enhanced photosynthesis. Furthermore, CU is also required for chlorophyll production [131], which might be needed when photosynthesis is enhanced. However, the reason for the upregulation of the CU transporter should be analyzed in further studies.

After 4 h of salt stress, the amount of cell membrane protein PIEZO is enhanced by 3.03 (Table 5). PIEZO is a non-selective cation channel mediating currents in response to mechanical stimulus. Furthermore, it is required for mechanical stress-induced Ca^2+^ signaling [132]. As salt stress leads to shrinking of the *Dunaliella* cells [5], this might lead to a mechanical stimulus, which increases the Ca^2+^ signaling and activates further downstream pathways.

As hypersaline conditions reduced the bioavailability of iron [133], it has been reported that transferrin was upregulated after hyperosmotic changes [134]. However, in our study, no proteins related to iron uptake were enhanced within 4 h of salt stress, such as transferrin or proteins from heme synthesis. Only the amount of ferredoxin, which is an iron–sulfur protein that mediates electron transfer, was found to increase (30 min: 2.15 and 2.19, 4 h: 3.08-fold) as well as the ferredoxin-thioredoxin-reductase (24 h: 1.97-fold) and ferrous iron transport protein B (24 h: 2.02-fold).

### 2.10. Further Adaption to Salt Stress and Comparison to D. salina

Gcn5-related N-acetyltransferase (GNAT) was found 1.66-fold upregulated at 24 h of salt stress. GCN5 is a highly prominent histone acetylase that is important for chromatin remodeling of promoters, facilitating the activation of transcription. The results of Li et al. [135] reveal that the enhanced expression of a GNAT resulted in a notable reduction in Na^+^ accumulation and oxidative damage. This effect was achieved by increasing the transporting activities of plasma membrane H^+^-ATPase and vacuolar H^+^-ATPase, which was also upregulated in our study.

In previous research, it was reported that certain proteins, such as tubulin, were upregulated during salt stress [16,17,18,19]. In our strain, tubulin levels were increased at 30 min of salt stress (3.54-fold). Additionally, the tubulin binding cofactor A (4 h: 1.79-fold) and the gamma-tubulin complex (24 h: 2.15-fold) displayed upregulation as a response to salt stress. *Dunaliella* cells shrink immediately when salt stress occurs, followed by a recovery of cell volume when sufficient glycerol is synthesized. Therefore, tubulin might lead to reorganization of the membrane skeleton structures and plasma membrane integrity. Previous studies reported that changes in the amount of cytoskeleton-associated proteins are related to an adaptive response during osmotic stress [136]. Furthermore, Livanos et al. [137] revealed the interplay between ROS and the tubulin cytoskeleton and reported how tubulin reorganization and remodeling occur in response to ROS homeostasis [136,137].

Compared to *D. salina*, our working strain *D. tertiolecta* lacks the significant β-carotene synthesis for which the *Dunaliella* genus is known. Furthermore, independently of the carotene production, in some species of the *Dunaliella* genus, high salt concentration forces the cell to accumulate and form cell clumps, which are classified as the palmella stage of the cell [26,138]. Algae enter the palmella stage under extreme conditions, such as a decrease or increase in salinity. Cells in the palmella stage become more circular and lose their flagella and eyespot [138]. This characteristic and the related proteins were also not present in our *D. tertiolecta* strain.

Nevertheless, the mechanism of salt tolerance is complex, and there are numerous unknown genes and proteins differentially expressed and synthesized during salt shock [139]. In this study, we identified several proteins in *D. tertiolecta* that were down- or upregulated after salt stress. The most relevant proteins we have identified in response to salt stress are summarized in Figure 5 and listed in Table 6. Even though common mechanisms exist in the salt stress response across diverse algae strains, such as the accumulation of osmolytes, there are significant differences between various strains, which might even be unique to specific species within one strain [140]. Many of our identified enzymes have already been associated with salt or oxidative stress in algae and in *D. salina*, such as the Ca^2+^ signaling pathway. However, some enzymes have not been identified by other studies analyzing *D. salina*’s adaption to salt stress and might be a unique characteristic of *D. tertiolecta*, such as CP12, ELMO, SnRK1, and galactokinase. ELMO might have a crucial function in the reorganization of the cell. SnRK1 is the plant equivalent of the heterotrimeric AMPK/Snf and mediates the crosstalk between ABA and salt stress signaling.

In our study, the amount of antioxidative enzymes and proteins involved in ROS and oxidative stress management were not differentially upregulated. CP12 plays a protective role against oxidative stress and suppresses the production of ROS, e.g., by controlling the electron flow from Photosystem I to NADPH. Consequently, it could serve as a substitute for the β-carotene production, which is observed in *D. salina* but not in *D. tertiolecta*.

Considering the upregulation of the carbonic anhydrases and CBC-related enzymes, while starch-degrading enzymes have not been upregulated, we suggest that glycerol synthesis mainly originates from photosynthesis. However, this might also be the case, as continuous light was applied for cultivation. Furthermore, the 11-fold upregulation of galactokinase after 30 min of salt stress suggests that glycerol synthesis might not only rely on photosynthesis and starch degradation but could also involve the degradation of other sugars like galactose. As the upregulation of galactokinase has not been found in *D. salina,* further research should compare the abundance and activity of this enzyme in *D. tertiolecta* with other *Dunaliella* strains.

In spite of the high quality of the genome, no sequence with adequate similarity to a typical G3pp could be identified. Either insufficient annotation leads to unsuccessful identification or the identified chimeric G3pdh might substitute the G3pp, as the chimeric G3pdh is able to convert DHAP directly into glycerol, leading to the quick production of glycerol [119]. If the G3pp is indeed absent and the chimeric G3pdh is responsible for the glycerol synthesis, this is a significant distinction between *D. tertiolecta* and *D. salina*, as G3pp has already been identified in *D. salina.*

For marine algae, NaCl is the primary salt leading to osmotic stress [141], therefore, this study focuses on the adaption of *D. tertiolecta* to increased NaCl concentration. As Gilmour et al. [142] demonstrated, alternative osmolytes such as KCl resulted in the same changes in photosynthesis as NaCl, we assume similar adaptions of our working strain when cells are exposed to osmotic shock produced by other salts and substances. These adaptions include glycerol production [5], as well as the alteration of the down- or upregulated proteins, and the activation of Ca^2+^ signaling [84].

The results of the molecular and phylogenetic analysis of Highfield et al. [143] revealed that *D. tertiolecta* is the most distantly related strain to the other evaluated *D. salina* strains within the *Dunaliella* genus. Additionally, in their study, they were able to classify their *Dunaliella* isolates to one of four sub-clades only, but they were not able to determine the exact species. Furthermore, Xu et al. [25] identified that their *D. salina* CCAP19/30 strain exhibits remarkable similarity to a strain of *D. tertiolecta* and hence, they assume that their *D. salina* in fact is a *D. tertiolecta.* This highlights the limitations in the exact identification of *Dunaliella* species only by phenotypic classification and analysis of phylogenic markers, such as ITS and 18S [144]. This problematic identification of *Dunaliella* strains hinders the information transfer from one *Dunaliella* strain to another, especially to new isolates. As a result, a comprehensive genome analysis becomes essential to unambiguously identify the *Dunaliella* species and enable strain-specific results. This will allow for a clear identification and classification of the specific strains being studied.

## 3. Materials and Methods

### 3.1. Algae Strains and Cultivation

*D. tertiolecta* UTEX 999 was acquired from the Culture Collection of Algae at the Georg August University of Göttingen (Göttingen, Germany). The microalgae were cultured in 500 mL Erlenmeyer flasks with a culture volume of 200 mL Johnson medium (pH 7.5) with 1 M NaCl [145]. The algae were cultivated in New Brunswick Innova 44 series shakers (Eppendorf, Hamburg, Germany) (28 °C, 120 rpm) equipped with light-emitting diodes (LED) from Future LED GmbH (Berlin, Germany) as described by Woortmann et al. [146]. Each flask was supplied with 2% *v*/*v* CO_2_ enriched air for aeration, which was regulated by a DASGIP^®^ MX module (Eppendorf AG, Hamburg, Germany). The microalgae were exposed to continuous illumination with a Photosynthetic Photon Flux Density (PPFD) of 100 μmol m^−2^ s^−1^, covering a spectrum between 400 nm and 750 nm, to mimic visible sunlight (color spectrum AM 1.5 G).

### 3.2. Phylogenetic Characterization of Strains

DNA extraction was performed with the InnuPrep plant DNA extraction kit (Analytic Jena AG, Jena, Germany, 845-KS-1060050). For identification, 18S rDNA was amplified using the primers EukA (21F) (AACCTGGTTGATCCTGCCAGT) and EukB (1791R) (GATCCTTCTGCAGGTTCACCTAC) [147]. Additionally, a 16S analysis was performed, to check for contamination using the 27F (AGAGTTTGATCATGGCTCAG) and 1492R (GGTACCTTGTTACGACTT) primer [148].

### 3.3. High Molecular Weight DNA Extraction and Library Preparation

Long-read sequencing was performed with high molecular weight genomic DNA (HMW gDNA). For the extraction of the HMW gDNA, a plant-optimized CTAB—PCI extraction method based on different protocols [149,150,151,152] was used; The CTAB extraction buffer (2% CTAB, 100 mM Tris pH 8.0, 20 mM EDTA, 1.4 M NaCl) was supplemented with 2% polyvinylpyrrolidone (PVP) and solved at 60 °C. Cells were harvested by centrifugation at 2500× *g* for 5 min; 0.5 g of fresh biomass was mixed with 10 mL buffer and incubated with 400 µL Proteinase K (Qiagen, Venlo, The Netherlands) for 30 min at 50 °C and was occasionally inverted. An amount of 2 mg RNAse A (Thermo Scientific, Waltham, MA, USA) was added and incubated for 10 min. The mixture was washed twice with one volume PCI (25:24:1) (10,000× *g*, 5 min, 10 °C), by saving and reusing the aqueous upper phase. Afterwards, the mixture was washed three times with chloroform. The HMW gDNA was pelleted by adding 30% PEG to the aqueous phase (1:4). The mixture was inverted, incubated for 30 min on ice, and centrifuged at 12,000× *g*, 10 °C for 30 min. The pellet was washed thrice with 70% ethanol (5000× *g*, 5 min, 10 °C). After the last step, samples were stored in ethanol at −20 °C until further use. To proceed, ethanol was removed (5000× *g*, 5 min, 10 °C), and the pellet was air-dried at 40 °C. After resuspension in 100 µL elution buffer (Qiagen, Venlo, The Netherlands), the size and quality of the HMW gDNA were analyzed by a Femto Pulse system (Agilent, Santa Clara, CA, USA), a Nanodrop photometer (Implen, Munich, Germany), and a Qubit dsDNA HS Kit (Thermo Scientific, Waltham, MA, USA), respectively. If the variation in DNA concentration between Nanodrop and Qubit was >50%, the DNA was purified with an electrophoretic clean up using a BluePippin system (Sage Science, Beverly, MA, USA) or an AMPure PB bead cleanup. For whole genome library preparation, 5 µg HMW gDNA were sheared in a gTube (Covaris, Woburn, MA, USA; 1700× *g*) by using the SMRTbell prep kit 3.0 (Pacific Biosciences, Menlo Park, CA, USA) according to the manufacturer’s instructions. For the size selection of the resulting library, AMPure PB beads were used. Libraries were stored at −20 °C. For sequencing, polymerase and primer were bound by using a Sequel II Binding Kit 3.2 (Pacific Biosciences, Menlo Park, CA, USA) according to the manufacturer’s instructions.

### 3.4. Total RNA Extraction and mRNA Sequencing

RNA extraction was performed with the RNeasy Plant Mini Kit (QIAGEN, Venlo, Germany) according to the manufacturer’s instructions. After extraction, RNA was treated with DNase using the TURBO DNA-free™ Kit (Thermo Scientific, Waltham, MA, USA). Depletion of ribosomal RNA, transcriptome library construction, and sequencing of the corresponding samples were performed by Eurofins Genomics (Ebersberg, Germany). Sequencing was conducted with the technology of Illumina NovaSeq 6000 with a 150 bp paired-end reading.

### 3.5. Genome Sequencing, Assembly, and Quality Assessment

A Sequel IIe (Pacific Biosciences, Menlo Park, CA, USA) was used for sequencing by applying 2 h pre-extension and 2 h adaptive loading (target p1 + p2 = 0.95) to an on-plate concentration of 85 pM, and 30 h movie time. For the initial de novo genome assembly, the SMRT Link (v11.0.0+, Pacific Biosciences, Menlo Park, CA, USA) was applied by using Improved Phased Assembly (IPA) [153]. The assembled sequences can be found within the National Center for Biotechnology Information (NCBI, NCBI BioSample accession number: SAMN37453276). To assess genome completeness, BUSCO, v5.3.2 (Benchmarking Universal Single-Copy Orthologs) was used [154].

### 3.6. Gene Prediction and Functional Analysis

Prediction of gene models was performed with Augustus-BRAKER, utilizing the assembled genome and mRNA short reads from NaCl-treated samples [31,32,33,34,35,36,37,38,39]. In addition, de novo prediction was performed using the existing Augustus training set for *D. salina* as well as the generated training set of *D. tertiolecta* at 1 M NaCl. The collected data were merged by removing redundant entries to create a comprehensive protein database using a custom Python script that eliminated identical protein sequences. Trinity (v2.13.1) [155] was used for mRNA assembly, and the coding sequences were predicted with CodAn (v1.1) [156]. Functional annotation was performed using DIAMOND (v20.0.15) [157] against the NR-NCBI (Dez. 2022) protein database. For functional characterization of exported protein sequences, KOALA (KEGG Orthology And Links Annotation [158], v3.0) and Mercator4 (v5.0) [159] were used. COG and GO terms were determined using EggNOG Mapper (v2.1.5) [160] and cateGOrizer (v3.218) [161]. If not further specified, default parameters were used for analysis.

### 3.7. Induction of Salt Stress

After 14 days of cultivation in 1 M NaCl, the NaCl concentration was abruptly increased to 2 M. For transcriptomic analysis, cells were harvested at 0 min, 30 min, and 4 h after adding NaCl. For proteomic analysis, cells were harvested after 30 min, 4 h, and 24 h after the increase in salt concentration. As a control, cells before the increase in NaCl concentration were used. Cells were harvested by centrifugation at 2500× *g* at 4 °C for 5 min.

### 3.8. Proteomics

#### 3.8.1. Protein Extraction

Cells were harvested before and 30 min, 4 h, and 24 h after changes in NaCl concentration (2500× *g*, 5 min, 4 °C). After cell harvest, samples were kept on ice, and all centrifugation steps were performed at 4 °C. Cell lysis and protein extraction were performed using the Protein Extraction Reagent Type 4 (Sigma-Aldrich, St. Louis, MO, USA) (1:3, *v*/*v*). For purification and precipitation, four times the volume of Methanol (with 10 mM Diethiothreitol (DTT)) was added to the samples. After vortexing, the initial volume of chloroform was added, and the mixture was vortexed again. Three times the volume of bidest. water was added to the mix. The mixture was centrifuged after an additional vortexing step (15,000× *g*, 1 min). The top aqueous layer was removed, and four times the volume of methanol (with 10 mM DTT) was added and vortexed. After centrifugation, the supernatant was removed, and the pallet was washed with acetone. The pellet was air-dried and dissolved in 8 M urea solution supplemented with 10 mM DTT. Three biological and three technical replicates were prepared for every condition.

#### 3.8.2. Protein Quantification and SDS-PAGE

Protein concentration was determined using a Nanophotometer (NanoPhotometer NP80, Implen GmbH, München, Germany) at 280 nm absorbance and with the Bradford assay [162]. Protein extracts were separated on a 12% one-dimensional SDS polyacrylamide gel for visual analysis of the qualitative variances in protein levels, according to Awad et al. [26].

#### 3.8.3. In-Gel Digestion of Protein Samples and LC-MS/MS Analysis

In-gel digestion of protein samples was carried out according to the method of Fuchs et al. [163]. However, instead of 10% Criterion™ Tris–HCl Protein Gel, a Short 12% SDS polyacrylamide gel was used. LC–MS/MS analysis, using a timsTOF Pro mass spectrometer equipped with a NanoElute LC System (Bruker Daltonik GmbH, Bremen, Germany) on an Aurora column (250 × 0.075 mm, 1.6 µm; IonOpticks, Hanover St., Australia), was carried out according to Engelhart-Straub et al. [164]. For measurement normalization, quality control samples were prepared by mixing 1 µL of each sample and analyzed at equal intervals between samples.

#### 3.8.4. Quantitative Analysis of Peptides and Proteins

Peptide and subsequent protein identification were performed using the PEAKS Studio software 10.6 (Bioinformatics Solutions Inc., Waterloo, ON, Canada) [165,166,167] and the own-designed genome database. Search parameters included a fragment ion of 0.05 Da and a precursor mass of 25 ppm using monoisotopic mass. As the digestion enzyme trypsin was selected, a maximum of two missed cleavages per peptide were allowed. The search was limited to at least one unique peptide per identified protein, and the false discovery rate (FDR) was set to 1.0%. The comparison of the different time points was performed with the Quantification tool PEAKSQ, with an Ion Mobility Tolerance of 0.05 Da, mass error tolerance of 20.0 ppm, and a Retention Time Shift Tolerance of 6 min (Auto Detect). Significance and fold change were set to 2, and all proteins were exported.

## 4. Conclusions

The cellular salt stress response in *Dunaliella* effects the biochemical metabolic pathways of the cell in a time-dependent manner. While the accumulation of compatible solutes such as glycerol is a commonly known adaptive strategy to maintain cellular function during salt stress, the regulation and the adaptation on the protein level have not yet been investigated in *D. tertiolecta*. Consequently, this study examined *D. tertiolecta*’s protein expression pattern after 30 min, 4 h, and 24 h, respectively. In contrast to *D. salina*, research on NaCl stress adaptation and, subsequently, genomics and proteomics research is limited in *D. tertiolecta*. To address this constraint, extensive genomic, transcriptomic, and proteomic analysis was performed allowing the identification of molecular mechanisms involved in NaCl stress adaptation. In this approach, mRNA sequences, in particular isolated from selected NaCl stress, allowed the generation of a unique and comprehensive set of protein sequences. The subsequent proteome analysis uncovers a greater number of proteins definitely expressed upon selected NaCl concentrations. With the establishment of *D. tertiolecta* as a reference, the path is opened to transfer the methods onto unspecified *Dunaliella* strains.

Proteomic analysis revealed a higher number of proteins upregulated than downregulated upon NaCl stress for all three time points. The majority of proteins with significant changes in abundance were categorized into energy production and conversion, amino acid transport and metabolism, and posttranslational modification. Furthermore, after 24 h of salt stress, a substantial upregulation of proteins involved in RNA processing and modification as well as translation, ribosomal structure, and biogenesis was observed.

Following salt stress, proteins associated with Ca^2+^ signaling, such as calcium/calmodulin-dependent protein kinases, glutamate receptors, and PIEZO, were upregulated. This indicates that Ca^2+^ signaling might be one of the initial responses of *Dunaliella* cells to salt stress. Within 30 min of salt stress, the voltage-dependent anion channel protein 2 exhibited enhanced abundance suggesting its requirement as a rapid adaption to salt stress. Additionally, sodium hydrogen exchange was observed to be phosphorylated only after 30 min of salt stress, which might indicate an activation of the enzyme. It is likely that these two enzymes play a crucial role in maintaining the cell’s ion homeostasis after salt stress.

Glycerol synthesis, which is required as an osmolyte to counteract high salt concentration, occurs via photosynthesis or starch degradation. While starch-degrading enzymes did not show any changes in abundance, we identified the upregulation of various proteins associated with photosynthesis, particularly after 30 min and 4 h of salt stress. Furthermore, carbonic anhydrases along with enzymes related to the CBC have been upregulated. These enhanced proteins indicate that in our study, glycerol synthesis mainly originates from photosynthesis, which might result from continuous light cultivation. Nevertheless, the substantial 11-fold upregulation of galactokinase after the initial 30 min of salt stress suggests that glycerol synthesis might not only rely on photosynthesis and starch degradation but could also involve other sugar degradation like galactose. Furthermore, the glycolysis-related protein PFK might be important for the long-term adaptation to high salt concentration.

In *Dunaliella*, the four enzymes, G3pdh, G3pp, Dhar, and Dhak, are involved in the glycerol synthesis and degradation pathway. However, our database did not enable the identification of G3pp. Nevertheless, we discovered a di-domain G3pdh isoform, which combines the conventional G3pdh domain with a phosphoserine phosphatase-like domain. This chimeric protein catalyzes the direct conversion of DHAP to glycerol, indicating that in our *D. tertiolecta* strain, the G3pp indeed might be absent and the phosphatase-like domain of the chimeric G3pdh converts the G3P to glycerol. If this is the case, and the G3pp is absent, this is a significant distinction between *D. salina* and *D. tertiolecta*, as G3pp has already been identified in *D. salina*.

None of the identified key enzymes of the glycerol synthesis pathway, including the chimeric G3pdh, were upregulated by 4 h after an increase in salt concentration, indicating their presence in sufficient quantities even before changes in extracellular salt concentration occurred. However, they might be activated through other protein modifications including the Ca^2+^ signaling pathways, which need to be analyzed in future studies.

However, despite the presence of common mechanisms in the salt stress response among various algae strains, such as the accumulation of osmolytes and ion homeostasis, there are significant distinctions not only between different strains but also within species [140]. We identified many proteins that exhibited either down- or upregulation following exposure to salt stress. However, some enzymes, such as CP12, ELMO, SnRK1, and galactokinase, have not been identified in previous studies examining *D. salina*’s adaptation to salt stress. It is likely that these enzymes are specific to *D. tertiolecta*. For instance, CP12 was upregulated at each analyzed timepoint and might serve as an alternative way for the protection against ROS, as we did not observe an increase in β-carotene or proteins related to β-carotene synthesis in our algae following salt stress. This indicates a potentially distinct adaptive response in *D. tertiolecta* compared to *D. salina*. However, this should be analyzed in more detail in further studies. Additionally, the alteration of oxidative stress with a special focus on ROS as a response to salt stress should be studied in more detail.

In conclusion, the proteome analysis on the basis of the own-designed database allowed for the identification of differentially expressed proteins involved in adaption to salt stress. Even though we did not find any upregulation of glycerol synthesis-involved proteins, we were able to identify a chimeric G3pdh that might have the capacity to enhance the glycerol yield as well as the tolerance of economically significant crops. Additionally, the other identified upregulated proteins might promote glycerol synthesis, e.g., by protecting the cell against ROS. Future studies could address these findings through process design interventions or genetic engineering to enhance glycerol production. Improved algae-based glycerol holds the potential for conversion into environmentally friendly “green chemicals”.

## Figures and Tables

**Figure 1 ijms-24-15374-f001:**
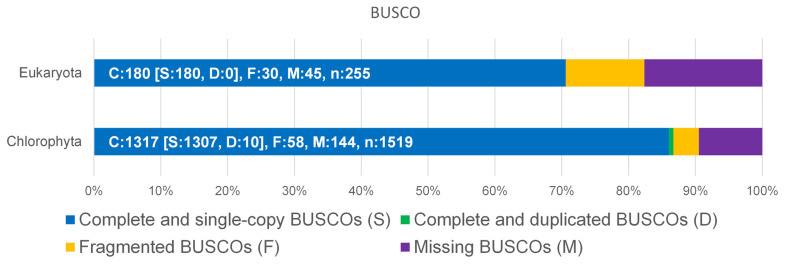
Benchmarking Universal Single-Copy Orthologs (BUSCO) was used for the assessment of the completeness of *D. tertiolecta* against Eukaryota and Chlorophyta datasets.

**Figure 2 ijms-24-15374-f002:**
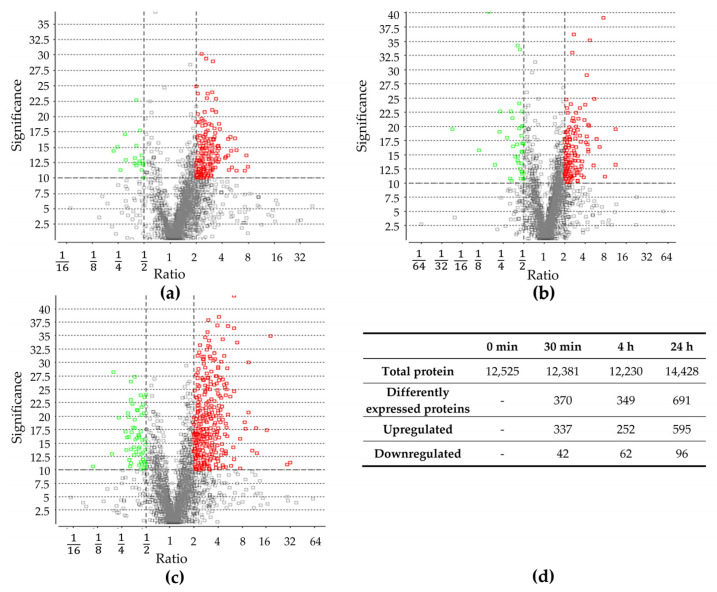
Differentially expressed proteins after (**a**) 30 min, (**b**) 4 h, and (**c**) 24 h of increase in salt concentration compared to proteins present before salt stress. Volcano plot of proteins, which were quantified with LC-MS/MS and PEAKS Studio software. Each point represents the difference in expression (fold change/ratio) plotted against the level of statistical significance. Proteins represented by a red and green circle exhibit significantly in- or decreased abundance, respectively. Grey circles did not exhibit significantly changed abundance. (**d**) Number of total identified proteins before and after 30 min, 4 h, and 24 h of salt stress; number of total differentially expressed proteins, as well as down- and upregulated ones after 30 min, 4 h, and 24 h of salt stress.

**Figure 3 ijms-24-15374-f003:**
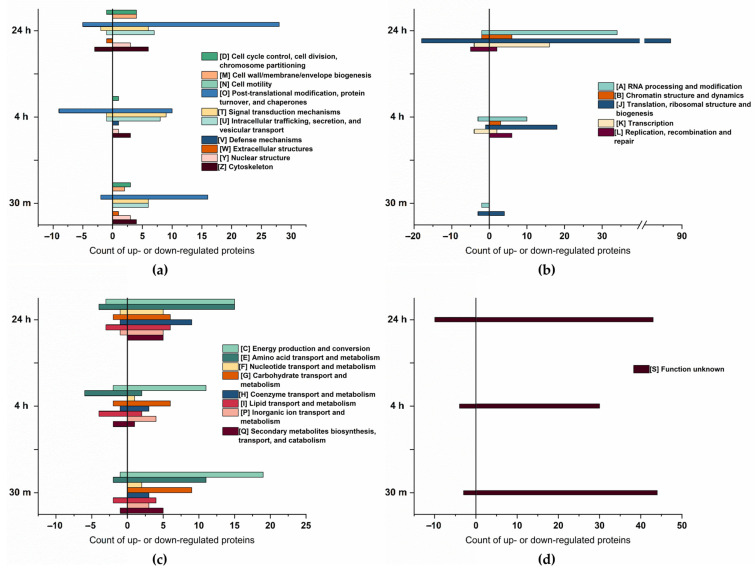
Cluster of Orthologous Groups (COG) classification of the differentially expressed proteins at 30 min, 4 h, and 24 h of salt stress. (**a**) COG of cellular process and signaling, total count. (**b**) COG of information storage processing, total count. (**c**) COG of metabolism, total count. (**d**) COG of poorly characterized proteins, total count.

**Figure 4 ijms-24-15374-f004:**
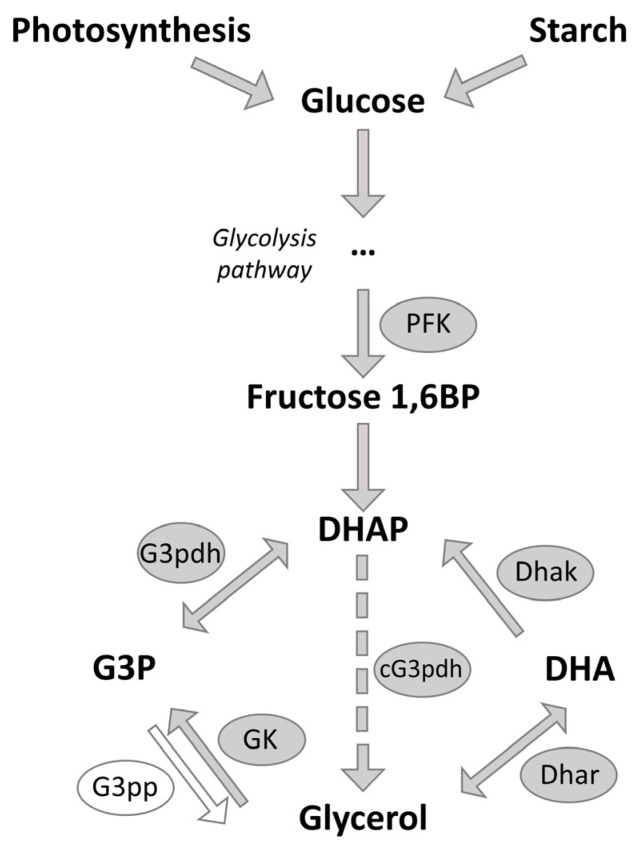
Schematic illustration of the glycerol synthesis pathway via photosynthesis or starch degradation. cG3pdh: chimeric glycerol-3-phosphate dehydrogenase; DHA: Dihydroxyacetone; Dhak: Dihydroxyacetone kinase; DHAP: Dihydroxyacetone phosphate; Dhar: Dihydroxyacetone reductase; Fructose 1,6BP: Fructose-1,6-bisphosphate; G3P: Glycerol-3-phosphate; G3pdh: Glycerol-3-phosphate dehydrogenase; G3pp: Glycerol-3-phosphate phosphatase; GK: Glycerol kinase; PFK: phosphofructokinase. The cG3pdh might substitute the absence of G3pp.

**Figure 5 ijms-24-15374-f005:**
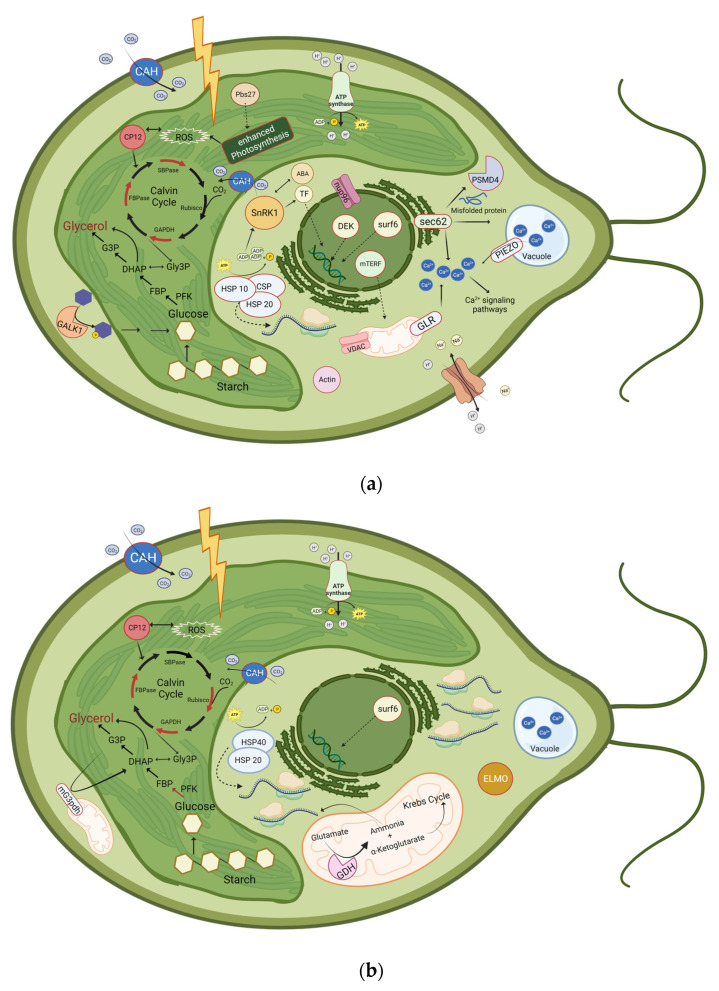
Schematic illustration of proteins that were significantly upregulated or phosphorylated at (**a**) 30 min and 4 h, or at (**b**) 24 h after induction of salt stress. Upregulated proteins are represented with a red border or as a red arrow. Downregulated enzymes are represented with a green arrow. ABA: Abscisic Acid; CAH: Carbonic anhydrase; CP12; CSP: Cold-shock protein; DHAP: Dihydroxyacetone phosphate; ELMO: Engulfment and cell motility protein; FBP: Fructose 1,6-bisphosphate; FBPase: Fructose-bisphosphatase; G3P: Glycerol-3-phosphate; GALK1: Galactokinase 1; GAPDH: Glyceraldehyde-3-phosphate dehydrogenase; GDH: Glutamate-dehydrogenase, Gly3P: Glyceraldehyde-3-Phosphate; GLR: glutamate receptor ; HSP10: Heat-shock protein 10; HSP20: Heat-shock protein 20; HSP 40: Heat-shock protein 40; mG3pdh: mitochondrial glycerol-3-phosphate dehydrogenase; mTERF: mitochondrial Transcription Termination Factors; nup96: Nuclear pore protein;Pbs27: Photosystem II Pbs27 protein; PFK, Phosphofructokinase; PSMD4: 26S proteasome non-ATPase regulatory subunit 4; ROS: Reactive oxygen species; Rubisco: Ribulose 1,5-bisphosphate carboxylase/oxygenase; SBPase: Sedoheptulose-bisphosphatase; SnRK1: SNF1-related protein kinase 1; surf6: Surfeit locus protein 6; TF: Transcription factor; VDAC: Voltage-dependent anion channel. Additionally, translation processes are highly enhanced after 24 h of salt stress, as 57 proteins related to ribosomes are upregulated. Even though the glycerol pathway is shown in the chloroplast, it additionally takes place in the cytosol. Created with BioRender.com.

**Table 1 ijms-24-15374-t001:** Proteins with the highest increase in abundance at 30 min, 4 h, and 24 h of salt stress.

Enzyme	30 min	4 h	24 h
galactokinase	11.46	3.30	-
ADP-ribosylation Crystallin J1. Source PGD	8.01	-	-
HAT (Half-A-TPR) repeats	7.85	3.60	-
Snf1-related protein kinase 1 (SnRK1)	5.56	-	-
nuclear protein 96	5.53	-	-
belongs to the actin family	4.93	-	-
translocation protein Sec62	4.88	-	-
rieske-like [2Fe-2S] domain	4.67	-	-
photosystem II Pbs27	-	11.05	-
sulfhydryl oxidase ALR/ERV	-	9.21	-
mitochondrial Transcription Termination Factors (MTERF) superfamily	-	7.43	-
thiamin pyrophosphokinase, vitamin B1 binding domain	-	6.40	-
EF hand family protein	-	4.61	-
adaptin C-terminal domain	-	4.45	-
CP12	2.50	4.25	2.48
thioredoxin-like domain	-	4.03	2.44
ELMO/CED-12 family	-	-	31.89
glutamate dehydrogenase	-	-	23.02
7,8-dihydroneopterin aldolase/epimerase/oxygenase	-	-	12.17
surfeit locus protein 6	3.63	-	12.09
ketohexokinase activity	-	-	9.45
biotin and thiamin Synthesis-associated domain	-	-	8.97
large subunit ribosomal protein L6	-	-	6.51
large subunit ribosomal protein L11e	-	-	6.17

**Table 2 ijms-24-15374-t002:** Proteins with the highest decrease in abundance after 30 min, 4 h, and 24 h of salt stress.

Enzyme	30 min	4 h	24 h
protein-l-isoaspartate O-methyltransferase	0.24	-	-
ribosomal protein L7 L12	0.24	0.31	-
RNA-binding protein with multiple splicing	0.32	-	0.29
belongs to the DEAD box helicase family	0.42	-	-
acyl carrier protein-like protein	0.44	-	0.25
AIG1 family	-	0.05	-
glutathione S-transferase	-	0.13	
phosphomannomutase	-	0.15	0.47
serine threonine-protein kinase	-	0.15	-
enoyl-(Acyl carrier protein) reductase	-	0.16	-
chloroplast envelope transporter	-	-	0.11
Isy1-like splicing family	-	-	0.12
Sugar (and other) transporter	-	-	0.20
photosystem ii reaction center w	-	-	0.31
sigma 54 modulation protein/S30EA ribosomal protein	-	-	0.31

**Table 3 ijms-24-15374-t003:** Differentially synthesized proteins related to photosynthesis at 30 min, 4 h, and 24 h of salt stress.

Enzyme	30 min	4 h	24 h
chlorophyll A-B binding protein	2.66	2.31	0.45
major light-harvesting chlorophyll a/b protein 3	3.3	-	2.02
photosystem II biogenesis protein Psp29	1.67	-	-
photosystem II subunit 28	2.21	-	-
photosystem I assembly	2.60	1.63	-
the polypeptide of the oxygen-evolving complex of photosystem II	2.27	-	-
photosystem I light-harvesting chlorophyll-a/b protein 2	-	1.75	-
plastocyanin/azurin family-domain-containing protein	-	3.10	2.80
photosystem II Pbs27	-	11.05	-
photosystem ii reaction center w	-	-	0.31

**Table 4 ijms-24-15374-t004:** Differentially synthesized proteins related to heat- and cold-shock proteins and chaperons after 30 min, 4 h, and 24 h of salt stress.

Enzyme	30 min	4 h	24 h
small molecular HSP 10	2.43	2.66	2.54
HSP20-like chaperone	3.04	-	-
cold-shock protein domain	2.05	-	-
HSP 70C	-	0.42	-
heat-shock chaperonin-binding motif	-	1.94	1.79
activator of Hsp90 ATPase	-	2.49	
zinc chaperone CobW-like	-	5.24	-
HSP20-like chaperone	-	-	2.24
DNAJ-like protein/HSP 40	-	-	2.88

**Table 5 ijms-24-15374-t005:** Differentially synthesized proteins related to channels and transporter after 30 min, 4 h, and 24 h of salt stress.

Enzyme	30 min	4 h	24 h
voltage-dependent anion channel protein 2	2.42	-	-
Ctr copper transporter family	-	2.34	2.27
potassium:proton antiporter activity	-	3.64	-
ZIP Zinc transporter	-	2.36	-
polycystin cation channel	-	2.09	-
PIEZO non-specific cation channel	-	3.03	-
chloroplast envelope transporter	-	-	0.11
major facilitator superfamily domain-containing protein	-	-	0.20

**Table 6 ijms-24-15374-t006:** Most relevant proteins related to salt stress adaption that were identified in this study. Name, function, and either their fold change or their phosphorylation (P) are presented after 30 min, 4 h, and 24 h of salt stress.

Enzyme	30 min	4 h	24 h	Function
glutamate receptor (GLR)	P	-	-	Ligand-binding and ion channel activity: Involved in Ca^2+^ signaling processes
sodium hydrogen exchanger	P	-	-	Intracellular regulation of pH and Na^+^ concentration
26S proteasome non-ATPase regulatory subunit 4 (PSMD4)	P	-	-	Protein degradation
galactokinase	11.46	-	-	Phosphorylation of galactose to galactose-1-phosphate
Snf1-like protein kinase (SnRK1)	5.56	-	-	Plant equivalent of the heterotrimeric AMPK/Snf. Mediates the crosstalk between ABA and salt stress signaling
PIEZO	-	3.03	-	Non-selective cation channel. Mechanical stress-induced Ca^2+^ signaling
calcium/calmodulin-dependent protein kinases	3.42	2.38, 4.61	0.37	Serine/threonine kinase activated by increasing Ca^2+^ concentration
CP12	2.50	4.25	2.48	Protecting cells against oxidative stress and suppressing the production of ROS. Control of electron flow from Photosystem I to NADPH
gamma carbonic anhydrase 3	1.75	-	-	CO_2_ uptake
carbonic anhydrase	-	-	1.87	CO_2_ uptake
fructose-bisphosphatase	2.34	-	2.93	Calvin–Benson cycle
sedoheptulose-bisphosphatase	3.66	-	-	
glyceraldehyde 3-phosphate	2.78	-	1.85	
ribulose 1,5-bisphosphate carboxylase/oxygenase (Rubisco)	-	-	2.95	
phosphofructokinase (PFK)	-	-	2.26	Conversion of glucose to fructose-1, 6-bisphosphate
engulfment and cell motility (ELMO) protein	-	-	31.89	Regulates cytoskeletal reorganization, cell apoptosis, and cell migration
glycerol-3-phosphate dehydrogenase (G3pdh) with di-domain	-	-	-	Direct conversion of DHAP to glycerol

## Data Availability

Not applicable.

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
