# Peer review of "The Time-Resolved Salt Stress Response of Dunaliella tertiolecta—A Comprehensive System Biology Perspective"

_ijms, 2023, doi:10.3390/ijms242015374_

Round 1
Reviewer 1 Report
Manuscript ID ijms-2649020-peer-review
The time-resolved salt stress response of Dunaliella tertiolecta - a comprehensive system biology perspective
Linda Keil, Norbert Mehlmer, Philipp Cavelius, Daniel Garbe, Martina Haack, Manfred Ritz, Dania Awad, Thomas Brück
The work is written competently. The study was well designed and technically sound.
Minor comments:
The abstract of the manuscript should include a reference to 24-hour light for algae under saline conditions. In this case, the possible regulatory role of light in the formation of salinity tolerance will be shown.
In a paper (Goyal, 2007) on the biosynthesis of glycerol from starch, microalgae were cultured under light and dark conditions (16/8 hours).
Page 2 Line 58 and 62: The authors used different abbreviations for the same enzyme. One abbreviation should be introduced.
Page 8 Line 297: If this is the coding enzyme, the SIT4 gene should be written in italics.
Page 10 Line 376: Please clarify the text. The text gives two values for RuBisCo at 24 h duration of stress exposure.
Page 10 Line 397: A typo in the name of the reducing agent molecule should be corrected. Replace "NADH" with "NADPH".
Conclusions: Cultivation conditions (24-hour lighting) of microalgae should be introduced in the text of the conclusions. The presence of a constant external energy source may play an important role in the regulation of microalgae adaptation responses to stress factor.

Author Response
x

Reviewer 2 Report
In this manuscript, Keil and co-workers present a laborious study regarding the response to salt stress of the unicellular green alga Dunaliella tertiolecta, using a time-resolved systems biology approach. The study covers relevant aspects such as genomic, transcriptomic and proteomic analysis of cells exposed to salt shock and identifies significant players involved in stress response and adaptation. The work is solid and precise, highlighting aspects which are clearly relevant for understanding the mechanisms involved in adaptation to salt stress. The manuscript is very well written, following a clear and logical pattern. In this reviewer’s eyes, there is only one minor issue that the authors need to address before the manuscript is accepted for publication. This issue is specified below.
- Glycerol production is a general response to hyperosmotic shock. The authors should comment on the possibility that similar response can be expected when cells are exposed to osmotic shock produced by other substances (including other salts).
Author Response
Reviewer 2:
In this manuscript, Keil and co-workers present a laborious study regarding the response to salt stress of the unicellular green alga Dunaliella tertiolecta, using a time-resolved systems biology approach. The study covers relevant aspects such as genomic, transcriptomic and proteomic analysis of cells exposed to salt shock and identifies significant players involved in stress response and adaptation. The work is solid and precise, highlighting aspects which are clearly relevant for understanding the mechanisms involved in adaptation to salt stress. The manuscript is very well written, following a clear and logical pattern. In this reviewer’s eyes, there is only one minor issue that the authors need to address before the manuscript is accepted for publication. This issue is specified below.
Glycerol production is a general response to hyperosmotic shock. The authors should comment on the possibility that similar response can be expected when cells are exposed to osmotic shock produced by other substances (including other salts).
The authors would like to thank the reviewer for the time spent on our manuscript. We included your mentioned issue in the text (page 17, lines 731-736). As marine algae are mainly affected by NaCl, this study focuses on changes in NaCl concentration. However, it is indeed worth mentioning, that other salts and substances might lead to similar adaption. KCl, for example was used in the publication Gilmour et al. (Gilmour et al. 1985) and results in similar changes in photosynthesis as NaCl.
Literaturverzeichnis
Gilmour, D. J.; Hipkins, M. F.; Webber, A. N.; Baker, N. R.; Boney, A. D. (1985): The effect of ionic stress on photosynthesis in Dunaliella tertiolecta : Chlorophyll fluorescence kinetics and spectral characteristics. In: Planta 163 (2), S. 250–256. DOI: 10.1007/bf00393515.
